# Physico-Mechanical and Durability Characterization of Eco-Ternary Cementitious Binder Containing Calcined Clay/Rice Husk Ash and Recycled Glass Powder

**DOI:** 10.3390/ma16217009

**Published:** 2023-11-01

**Authors:** Philbert Nshimiyimana, Ulrich Franck Tameghe, Christian Ramadji, Elodie Prud’homme, Zengfeng Zhao, Désiré Compaoré, Adamah Messan

**Affiliations:** 1Laboratoire Eco-Matériaux et Habitats Durables (LEMHaD), Institut International d’Ingénierie de l’Eau et de l’Environnement (Institut 2iE), Rue de la Science, Ouagadougou 01 BP 594, Burkina Faso; 2Groupe Hydro-Géotechnique Afrique, Cité Angré Les Manguiers, Cocody, Abidjan 00225, Côte d’Ivoire; 3MATEIS, Matériaux: Ingénierie et Science, INSA Lyon, Université de Lyon, 7, Avenue Jean Capelle, CEDEX, 69621 Villeurbanne, France; 4Department of Structural Engineering, College of Civil Engineering, Tongji University, 1239 Siping Road, Shanghai 200092, China; zengfengzhao@tongji.edu.cn; 5CIMBURKINA, Zone Industrielle de Kossodo, Ouagadougou 01 BP 6337, Burkina Faso

**Keywords:** cement, pozzolanic reaction, calcined clay, rice husk ash, recycled glass powder, mechanical property, durability

## Abstract

The objective of this study is to determine the influence of recycled glass powder (GP) on the physico-mechanical behavior and durability of a ternary cementitious binder containing calcined clay_metakaolin (MK) or rice husk ash (RHA). Different mortars were produced and characterized in fresh and hardened states. Reference mortars were produced using 100% cement CEM II/B-L 42.5R and 70% CEM + 30% MK or RHA. Test mortars were produced with the substitution of the MK or RHA with the GP and keeping the rate of the substitution at 30%; i.e., in ratios of 20:10, 15:15, and 20:10 of MK/RHA:GP. The water/binder weight ratio was maintained at 0.5, and the consistency of all mortars was adjusted using an admixture (superplasticizer/binder weight ratio of 0.75%). The substitution of MK and RHA with GP reduces the water demand to achieve the normal consistency of pastes and therefore increases the workability of mortars containing both binders CEM+MK+GP and CEM+RHA+GP. The substitution of MK and RHA with GP slightly reduces the compressive strength for both binders. The water-accessible porosity slightly increases for the substitution of MK and reduces for the substitution of RHA with GP. The mass losses after acid attack slightly increase with the substitution with GP, lower for the MK than the RHA up to 15% GP, but it remained far below that of 100% CEM. The results show that the substitution of MK and RHA with GP can improve the physical properties and durability of the mortars compared with that of 100% CEM, but it slightly decreases the mechanical properties due to the low rate of the pozzolanic reactivity of the GP. Further studies should seek to understand the reactivity behavior of the GP at the microstructure scale and therefore improve the mechanical performance of GP based mortar.

## 1. Introduction

Environmental problems are partly related to cement manufacturing due to the high demand for construction. This leads to the intensive consumption of energy and natural resources and high emissions of greenhouse gases. In fact, cement production contributes around 5–8% of global CO_2_ emissions [1]. This incites the cement industry, and the construction sector, to comply with the sustainable development goals that aim at making cities and communities sustainable and ensuring sustainable consumption of resources [2]. Therefore, the construction sector has been using alternative materials and systems [3,4,5]. 

More specifically, one of the approaches aims to replace clinker, the main constituent of ordinary Portland cement (OPC), with mineral additions with low energy and environmental impacts. Several studies have looked at the design of binary cementitious binder by substituting cement with calcined kaolinite clay, MK [6,7], Tuff [8], granite powder [9], and rice husk ash, RHA [10]. Other studies have focused on the recovery of glass waste as a partial replacement for cement [1,11,12,13]. 

Ntimugura et al. [6] showed that the substitution of OPC with 25% MK improved the compressive strength of mortars cured for up to 90 days and the resistance to acid attacks, among other parameters. Mounika et al. [10] showed that the substitution of cement with 30% RHA improved the mechanical and durability properties such as the compressive strength and resistance to acid attack of concrete. Sadiqul Islam et al. [14] showed that the substitution of cement with 20% ground glass resulted in acceptable results on mortar cured for up to 1 year. Ren et al. [15] found that the substitution of OPC with 65% of blast furnace slag led to a higher resistance toward acid attacks compared with the control. 

These studies show the potential to use materials from alternative resources to improve the physical, mechanical, durability, and environmental performances of cementitious binders. They showed that the substitution of cement with mineral additions achieves better performance at a content of 10 to 30% MK [6] and 20% RHA [10]. Nevertheless, these studies were limited to the design of binary binders and/or the use of additions obtained from an energy-intensive calcination process. Therefore, more studies can still look at the design of ternary binders for a better improvement in the overall performance of composite cement. This would potentially reduce the energy and carbon footprint due to calcination and add value to alternative cementitious additions.

The present study therefore aims at designing a ternary cementitious binder incorporating cement and two additives, essentially MK/RHA and GP: cement+MK+GP and cement+RHA+GP; to assess the feasibility of recovering GP in the production of cement. This would also make it possible to limit the use of MK and RHA, which are essentially produced through energy-intensive processes by the calcination of kaolinite clay and rice husk. It would also promote the use of GP, whose physical treatment is only based on grinding/sieving. The performances of the ternary binder were tested on the physical, mechanical, and durability behaviors of pastes and mortars in fresh and/or hardened states.

## 2. Materials and Methods

### 2.1. Materials

Cement is a CEM II/B-L 42.5R, produced by CIMBURKINA, according to EN 197-1 [16]. Calcined clay “metakaolin” (MK) is obtained by calcining, at 700 °C for 3 h, the kaolinite-rich clay collected from the locality of Saaba in Burkina Faso, after grinding and sieving under 80 µm. Table 1 presents the chemical composition of MK, containing mainly silica (57.9% SiO_2_) and alumina (38.3% Al_2_O_3_), according to a previous study in the laboratory [6].

Rice husk ash (RHA) is produced by calcining rice husk at 550 °C for 4 h, followed by cooling and grinding/sieving under 80 µm. RHA essentially contains 91.2% SiO_2_, according to a previous study in the laboratory [17]. The glass powder (GP) is produced by washing/drying pieces of broken white glass recovered from public dumps, followed by grinding and sieving on 80 µm. The GP essentially contains 68.1% SiO_2_, 14.5% CaO, and 12.2% Na_2_O, according to [14]. 

Figure 1a shows that the particle size distribution (PSD) of GP is comparable to that of CEM, at mainly around 60 µm, with a multimodal particle distribution. It also presents the PSD of MK and RHA.

The GP used in the present study fixes up to 1385 ± 56 mg/g of lime (Ca(OH)_2_), which is slightly lower than that of reactive calcined clays (from Saaba), which fixed 1600 mg/g of lime [18], and beyond the threshold of 630 mg/g required for pozzolanic clays [19]. However, its amorphicity content, measured using acidic dissolution referring to the method in [18], of 57.6 ± 1.1% is less than 70% of the most reactive calcined clay [18].

River sand was used to make mortars after washing and drying. The particle size analysis showed that the sand has a uniformity coefficient of 2.4 with a curvature coefficient of 0.75 for well-spread sand (Figure 1b). The superplasticizer type Sika Viscocrete was used in the mortars; it has a density of 1150 kg/m^3^. 

### 2.2. Mix Design and Sample Preparation 

Mortars are prepared by mixing sand, a binder containing different additions (MK or RHA and GP), and water. The water-to-binder weight ratio (W/B) is maintained at 0.5 and the sand-to-binder (S/B) at 3 for all mortars. The consistency of the mortars was corrected by adding a superplasticizer (Sp/B = 0.75%) to avoid the demand for more water from some additions, which would unnecessarily increase the porosity. Three (03) control designs: C100, C70M30, and C70R30 respectively containing 100% CEM, 70% CEM + 30% MK, and 70% CEM + 30% RHA were produced. Two sets of three (03) test designs were produced, each containing GP in partial substitution to MK or RHA but keeping the content of total substitution to 30%; i.e., in ratios 20:10, 15:15, and 10:20 of MK/RHA:GP. Table 2 presents the quantities of the constituents used for the design of each mixture.

The dry materials (sand+binders) were mixed for 2 min, then water containing the superplasticizer was added and mixed for 2 more minutes, using a Controlab 10031.5 automatic mixer, to produce mortars. The mortars were cast in a prismatic mold 40 × 40 × 160 mm^3^ and compacted using a standard 65-L0012/E control shock table machine. The mortars were covered with plastic bags to prevent water loss for 24 h. They were removed from the mold and immediately cured in water under controlled conditions (20 ± 1 °C) until the test ages of 2, 7, 28, and 90 days for the mechanical tests and 28 days for durability tests.

### 2.3. Samples Characterization

The pastes were also produced using only binders and water and used to immediately measure the water demand for achieving the normal consistency and the setting time at the normal consistency. The workability of fresh mortars was also measured by slump using a mini-cone.

The hardened mortars were tested in compression to assess their compressive strength on the two halves of each sample, using a hydraulic press, which has a capacity of 250 kN, at a loading speed of 50 ± 10 N/s until failure [20]. The values of resistance to compression, R_c_ (MPa), were deduced from Equation (1), where F (N) is the maximum load at break and b (40 mm) is the width.
(1)$$R_{c}=\frac{F}{b^{2}}$$
(2)$$C_{a}=\frac{Mx-Mdry}{A}$$
(3)$$E=\frac{M_{air}-M_{dry}}{M_{air}-M_{dry}}\times 100$$

The capillary absorption was tested on mortars, referring to [9], on samples previously dried at 60 °C in an oven to constant mass, placed in water at a height of 5 ± 1 mm to allow unidirectional water to rise from the bottom of the specimens. The evolution of the mass of absorbed water was recorded over time: 0.25, 0.5, 1, 2, 8, and 24 h. The capillary absorption coefficient, $C_{a}$ (kg/m^2^), was calculated using Equation (2), where Mx (kg) is the mass of the specimen at a given absorption time, M_dry_ (kg) is the initial mass of the dry specimen, and A (4 × 16 cm^2^) is the cross-section area of absorption of the specimen. The water-accessible porosity E(%) was measured by hydrostatic weighing on samples saturated in water using Equation (3). M_air_ and M_water_ are the masses of a saturated specimen weighed, respectively, in air and in water.

The resistance to acid attack was tested by measuring the variation in mass over the time of immersion in an acidic solution. The test specimens of mortar cured at 28 days were placed 5 cm under a solution of sulfuric acid (5% H_2_SO_4_ of the volume of water); then, the residual masses of mortar were measured after 2, 7, 14, and 28 days of exposure. The degree of acidity in the solution was adjusted regularly by pH measurement carried out every 7 days [9].

## 3. Results and Discussion

### 3.1. Physical Characteristics of Fresh Pastes and Mortars

#### 3.1.1. Workability

The workability was assessed based on the evolution of the water demand (W/B) for achieving the normal consistency of pastes and the slump of mortars. Figure 2a presents the evolution of W/B of the paste of cement containing MK and substituted with GP.

The results show that the paste containing 100% CEM has a W/B of 30%, while that containing 70% CEM and 30% MK has a W/B of 38.5%; i.e., an increase of 28% [(38.5 − 30)/30]. This confirms that the addition of MK decreases the workability of cement paste due to its water affinity. For different substitutions of GP in 30% MK, there is a decrease in W/B from 38.5% to 33% with 0 to 20% GP, i.e., a decrease of 14%. This shows that the GP improves the workability of the binder (CEM + MK), which can be related to its low affinity to water due to its non-porous character. Figure 2b presents the evolution of the W/B of cement containing RHA and GP: 30% RHA requires twice the W/B (60%) than 100% CEM (30%). The substitution of RHA with GP also decreases the W/B from 60 to 40% for 0 to 20% of GP, i.e., a decrease of 33%.

The increase in water demand (W/B) of cement with RHA is explained by its affinity to water due to its high porosity. Sisman et al. [21] and Mitrovi [22] also confirmed this character of RHA and MK, requiring a large amount of water to reach normal consistency. The presence of GP can be beneficial in terms of reducing this water demand, thereby counteracting the detrimental effect that would occur on the physico-mechanical performances of the binders and potentially improving their durability. While the MK and RHA substitution in cement resulted in a high increase in the water demand, the GP substitution in MK or RHA considerably reduced this water demand, respectively, by 14% and 33% to reach a similar consistency. This beneficial effect is more noticeable in RHA than in MK. These results confirm the non-porous character and the surface smoothness of the GP, as previously reported by Arab et al. [23].

Figure 3a presents the evolution of the slump from 8.5 cm for 100% CEM to 1.5 cm for 70% CEM + 30% MK.

This finding clearly shows the reduction in the workability of MK-based mortars. The slump increased from 1.5 cm to 7.5 cm when MK was substituted with 20% GP. This shows that glass powder increases the workability of MK mortars. Figure 3b also shows the reduction in the workability of mortars with RHA with a slump of 0.5 cm for 70% CEM + 30% RHA from 8.5 cm for 100% CEM. This value increased from 0.5 cm to 6.3 cm when RHA was substituted with 20% GP. This again shows that the GP increases the workability of RHA mortars. However, the influence of GP is more noticeable in MK than in RHA, mainly due to the high water demand of RHA compared with MK.

These results show that the GP has the effect of improving the workability of mortars containing MK and RHA. Previous studies have also demonstrated that GP would increase the slump of the mortar due to the difficult hydration character of glass particles and its smooth surface and low water absorption, i.e., waterproof character [13,23,24].

#### 3.1.2. Setting Time

Figure 4a presents the evolution setting time of the pastes of cement substituted with MK and GP.

The initial setting time decreases from 182 min for 100% CEM to 139 min for 70% CEM + 30% MK, unlike the final setting time, which increases from 245 min to 269 min. This can be related to the opposite effects between the increase in the setting time due to the increase in water demand to reach normal consistency and the decrease in the setting time due to the pozzolanic effect when MK was used in substitution with the cement. The substitution of MK with 10% GP increases the initial and final setting time up to 227 and 333 min, respectively, beyond which they were stable. Figure 4b presents the evolution of the setting time of cement substituted with RHA and GP. The initial setting time increases from 182 min to 243 min as well as the final setting time from 245 min to 330 min, respectively, for 100% CEM and 70% CEM + 30% RHA. The substitution of RHA with GP further increases the initial and final setting time up to 368 and 463 min, respectively, with 10% GP, beyond which they tend to decrease.

The results show that RHA increases the initial setting time, unlike MK. This can be justified by a higher amount of water required for RHA to reach normal consistency, which delays the physico-chemical phenomenon of setting. Mounika et al. [10] also reported the same character of RHA that increased setting times; however, Mitrovi [22] reported that MK reduced the setting times. The GP decreases the setting times of mortar, with a much more noticeable effect observed on RHA than on MK. This is attributed to the low water demand of the GP, thus facilitating the setting phenomenon. Therefore, the GP would contribute to limiting the risks of rapid shrinkage when it is used for work in hot weather or in desert areas, due to its low water demand and rapid setting.

### 3.2. Mechanical Characteristics and Durability Indicators of Hardened Mortars

#### 3.2.1. Compressive Strength

Figure 5a shows that the compressive strength of the reference mortar (100% CEM: C100) increased from 21.5 MPa at 2 days to 55.5 MPa at 90 days. This is slightly higher than the strength of mortar 70% CEM + 30% MK (C70M30), which evolved from 15.1 to 52.9 MPa from 2 to 90 days, i.e., a decrease of 30% [(15.1 − 21.5) × 100/21.5] at 2 days and only 5% [(52.9 − 55.5) × 100/55.5] at 90 days from the reference mortars. This confirms the contribution of metakaolin to the development of the strength of the cementitious matrix at a later age but remains significantly lower than the reference cement at an earlier age. This is due to the high content of MK (30%) and its slow pozzolanic reactivity at an early age. However, the strength at 7 days of C70M30 (30.3 MPa) was slightly higher than that of C100 (25.3 MPa), which can probably be related to the filler effect of MK. 

The substitution of 30% MK with 10, 15, and 20% GP gave quasi-equivalent values of the compressive strength, respectively, of 16.1, 15, and 15.1 MPa at 2 days, which relatively decreased to 44.9, 41.9, and 42.4 MPa at 90 days, i.e., −15%, −21%, and −20%. This shows a similar physical effect of MK and GP on the compressive strength at an early age and better reactivity of MK at a later age. Although the strength of mortars containing GP continuously increased with the maturation time, it did not catch up with that of MK-mortars until the 90th day. This is due to the slower pozzolanic reactivity of GP than MK, which is responsible for the development of the mechanical strength over time. Arab et al. [23] showed a similar evolution in the strength of cement containing 10% substitution of GP with values ranging from 68.7 MPa at 28 days to 81.4 MPa at 90 days.

Figure 5b shows that the compressive strength of the mortar containing 30% RHA (C70R30) is 15.2 MPa, which is also lower than 21.5 MPa of the reference mortar (C100) at 2 days. However, these values respectively increase to 61.5 MPa and 55.5 MPa at 90 days, i.e., an increase of 11% [(61.5 − 55.5) × 100/55.5] for RHA mortar from the reference mortar, which is attributed to the high pozzolanic reactivity of RHA. The substitution of RHA with 10, 15, and 20% GP, respectively, decreased the compressive strength to 13.2, 12.4, and 12.7 MPa at 2 days, i.e., −13%, −18%, and −16% and to 51, 48.7, and 41.1 MPa at 90 days, i.e., −17%, −21%, and −33%.

This shows that the substitution with 10% GP gives better results in MK or RHA than 15 and 20% GP. Jochem et al. [25] obtained values of 82 MPa at 90 days for the same substitution and with a particle size of <80 µm. Shao et al. [26] also showed that the mechanical activity indices of mortars could exceed 90% at a young age and 108% at 90 days with the substitution of cement with 30% GP of a grain size less than 38 µm. This shows that the compressive strength of the specimens containing GP is strongly influenced by its content and size. In addition, Esmaeili et al. [27] showed that strength depends on the parameter of the color of the glass, where glasses of green colors showed higher pozzolanic activity than others such as the white-colored GP used in the present study.

After evaluation of the effect of GP on the cementitious binders containing MK/RHA, it appears that the compressive strength of mortars increases with the time of maturation for all mixtures. However, it remains slightly below that of the reference. It also appears that RHA is more beneficial for long-term (90 days) strength development, whereas MK is better for improving the strength around 28 days. This can essentially be related to the early filler effect in addition to the pozzolanic effect of MK compared with the late pozzolanic effect of RHA. Subsequently, the effect of GP was analyzed on the durability indicators of this ternary cement, based on tests of water-accessible porosity, capillary absorption, and resistance to acid attack.

#### 3.2.2. Water-Accessible Porosity

Figure 6a presents the evolution of water-accessible porosity of MK- and GP-based mortars after curing for 28 days.

The results show that the evolution of the water-accessible porosity (11–12.6%) of mortars containing MK and 10–20% GP remains lower than that of 100% CEM (15.8%) and slightly higher than cement containing 30% MK (10.8%). This could be explained by the refinement in the pores following the filler effect and possibly increased pozzolanic cementitious products. It can also be related to the non-porous, non-absorbing, and smooth surface of GP, which reduced the water demand during the casting of mortars and achieved a high packing factor, reducing the porosity and water absorption after the curing/drying. Elaqra et al. [24] explained this character referring to the W/B ratio, which was constant to produce all mortars. The substitution of MK with GP in the matrix leads to the circulation of a large quantity of free water in the pores.

Figure 6b shows that the evolution of the water-accessible porosity (13.9–14%) of mortars containing RHA and 10–20% GP remains lower than that of 100% CEM (15.8%) and 70% CEM + 30% RHA (16.2%). It also shows that the water-accessible porosity of 30% RHA (16.2%) is much higher than that of 30% MK (10.8%). This suggests that the mortar containing RHA is more porous than that containing MK. This can be explained by the high-water demand of the mortar containing RHA (W/B = 0.60) compared with that of MK (W/B = 0.39) to reach normal consistency of their respective pastes, while the mortars were produced using W/B = 0.5. The W/B ratio decreases with GP substitution, with a higher decrease observed with RHA than with MK. Therefore, the substitution of RHA with GP decreased the porosity, relating to a better compacity of the mortar. This is due to the improved workability (increase of the slump) observed when the RHA was substituted with GP while maintaining the same W/B ratio of 0.5. This suggests that the decrease in the content of RHA in the mortar would reduce the circulation of free water as the water-accessible porosity tends to decrease, although its values remain higher than that of MK mortars.

#### 3.2.3. Capillary Absorption

Figure 7 presents the evolution of the coefficient of capillary water absorption with the square root of time, between 1 h and 24 h, of the mortars containing MK and RHA substituted with GP.

The results show that the absorption rate increased slightly with 30% MK and largely with 30% RHA and decreased with the substitution with GP in both cases. Apart from the C70M30, C70R30, and C70R20P10 mortars, the capillary absorption rate of different mortars is relatively lower than that of the control (C100). This agrees with previous studies on the substitution of cement with metakaolin [6,28]. It also agrees with the results obtained for the substitution of cement with GP [13]. This decrease in absorption is attributed to the pozzolanic effect in addition to the filler effect of the fine GP on the compacity, which leads to the densification of the microstructure. 

Table 3 summarizes the “sorptivity” the slopes of the coefficient of capillary absorption over the square root of time (Figure 7), which represents the rate of water absorption in the capillary pores.

It is observed that the values of the sorptivity decrease with the substitution with GP and more significantly in MK than in RHA due to the high affinity of RHA-rich mortars to water. This qualitatively suggests that the capillary pores decrease with GP substitution in the present ternary binders. It also agrees with the overall decrease in water absorption by total immersion after the substitution of MK and RHA with GP (Table 3). This is attributed to the filler effect of GP and eventually the formation of CSH by pozzolanic hydrates [10]. It is therefore an interesting indicator of the improvement in the durability of mortars by adding value to the GP in the ternary binder. 

#### 3.2.4. Resistance to Acid Attack

Figure 8a and b respectively show that the mass losses of mortars containing MK and RHA substituted with GP increased over the time of exposure to acid for all specimens. However, the mortars C70M30 and C70R30 lose less mass at all exposure times compared with the reference C100.

Nevertheless, the mass loss slightly increased with various GP substitutions, where was lower for MK than RHA by up to 15% GP, but beyond this, it kept increasing for MK and decreased for RHA. After 28 days of exposure, the C100 mortar lost 58.2% of its mass, while the C70M30 and C70R30 mortars lost only 7.7% and 18.4%, respectively. Nevertheless, the substitution of MK and RHA slightly increased the mass loss up to 18.9% with 20% GP and 24% with 15% GP, respectively. The mass loss is attributable to the dissolution of the binder hydration products in the acidic solution and is eventually followed by the loss of particles [15]. In fact, the mass loss would depend on the quality and maturity of the hydration products and increase with the water-accessible porosity. Ren et al. [15] reported that the substitution of cement with 65% of blast furnace slag decreases the mass loss due to acid attack compared with a control. Ramadji et al. [9] also reported that mortars containing granite powder were less sensitive to acid attack than a control. The authors also reported that the same mortars were more sensitive when they were cured for 28 days than for 90 days. This can be related to the pozzolanic consumption of portlandite (Ca(OH_2_)) from the hydration of cement with the mineral additions, which is also more effective after long-term curing. Moreover, Goyal et al. [28] reported that mineral additions lowered the detrimental effect of acid attack on concrete, which was more significant in ternary system cement–silica fume-fly ash than in binary system of cement–silica fume. This was related to the peeling of the surface layer due to continuous volumetric expansion of gypsum and ettringite respectively formed from the reaction of portlandite and calcium aluminate hydrate with sulfuric acid. Additionally, RHA was more beneficial for the long-term (90 days) development of the mechanical resistance and would eventually provide the same benefit for the resistance to acid attack. In contrast, MK was better for early (28 days) mechanical and acid attack resistance, which was related to the early filler effect in addition to the pozzolanic effect of MK compared with the pozzolanic effect of RHA. The results from the present study generally show that the mortars containing RHA have lower resistance to acid attack than those with MK, up to 15% GP.

This agrees with the previous studies that attributed the resistance to acid attack to the consumption of the soluble phase of portlandite by silica and alumina in the MK through the pozzolanic reaction [6]. Mounika et al. [10] have additionally attributed the resistance to acid attack to the resulting reduction in pore size, which makes the pore structure of mortar less accessible and less prone to acid attack. The substitution with GP slightly reduced the resistance to acid attack. This can be attributed to the fact that mortar containing GP may not have been fully matured at 28 days because GP is less reactive and would give better performance against long-term acid attacks. However, the GP substitutions show remarkably better resistance to acid attacks than 100% cement.

## 4. Conclusions

This study aimed to design a ternary binder incorporating metakaolin (MK)/rice husk ash (RHA) and recycled glass powder (GP) that can be used to produce eco-friendly cementitious binder with various effects on different physico-mechanical properties and durability. The following specific conclusions can be drawn:The substitution with GP in MK/RHA decreases the water demand (W/B) to reach the normal consistency of the paste. The W/B is 33% and 40% for the paste containing 20% GP in the substitution of MK and RHA, respectively. This is lower compared with the W/B of 38.5% and 60%, respectively, for the paste containing only 30% MK and RHA. This reduction in water demand would therefore improve the workability. Nevertheless, GP increases the setting time of the paste essentially due to its dilution effect, which would be beneficial for work in hot regions.The substitution with GP in MK/RHA did not sufficiently develop the compressive strength at an early age, displaying values lower than that of the controls. However, the strength relatively improved over time and significantly reduced the deficiency at 90 days.The substitution with GP in MK/RHA improved some of the durability indicators of mortars. It reduced the porosity and sorptivity of various mortars, with a more significant effect observed for RHA than MK. The mass loss after exposure to acid attack was also reduced with respect to cement, with a more significant effect observed with MK than RHA, up to 15% GP.

To fully understand the feasibility of GP in ternary cement, further analyses of the microstructure to better understand the chemical interactions, the change in formed products, and the evolution of the resistance are important and should be conducted. In addition, the thermal stability and resistance to alkali reactions of such binders, as well as their durability assessed at later age of curing (>28 days), should be considered.

## Figures and Tables

**Figure 1 materials-16-07009-f001:**
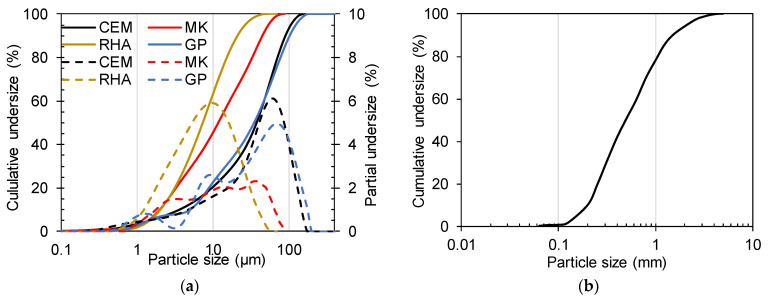
Particle size distribution of materials: (**a**) CEM, MK, RHA, and GP tested using Laser diffraction operated in humid mode and (**b**) sand tested by dry sieving.

**Figure 2 materials-16-07009-f002:**
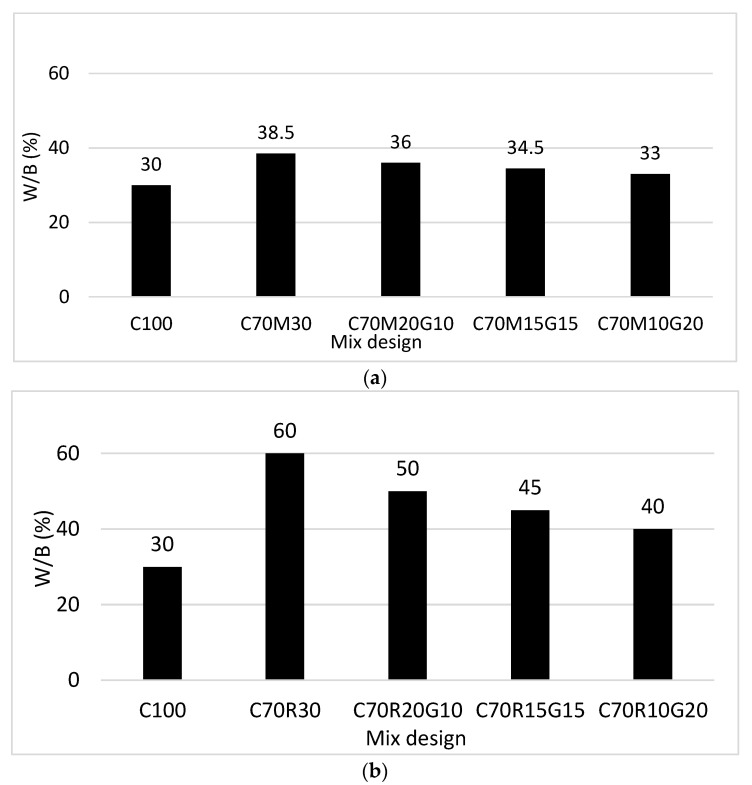
Water demand (W/B) to reach the normal consistency of binder pastes with different contents of glass powder (G) for binder containing: (**a**) metakaolin (M) and (**b**) rice husk ash (R).

**Figure 3 materials-16-07009-f003:**
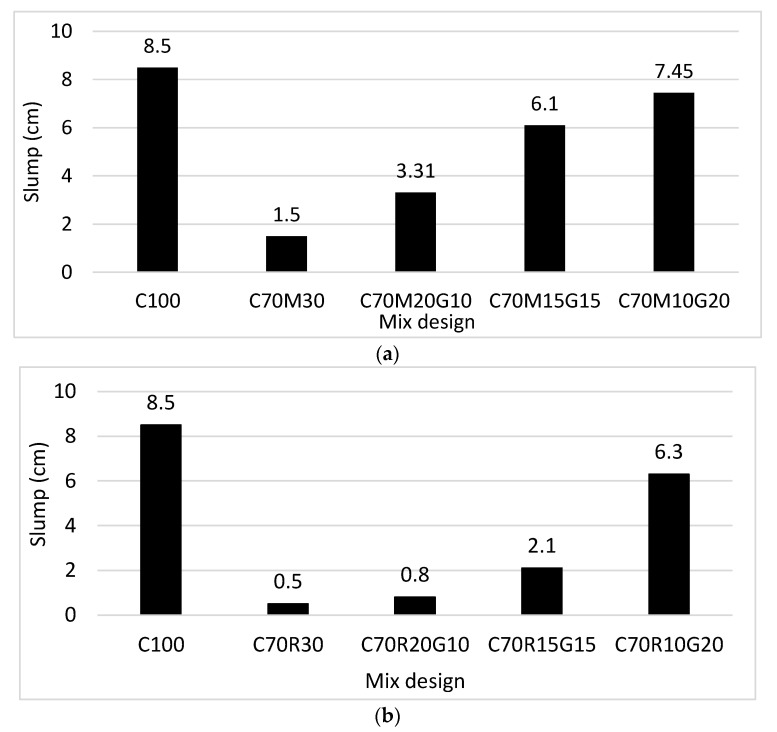
Slump on mini-cone of binder mortars with different contents of glass powder (G) for binders containing: (**a**) metakaolin (M) and (**b**) rice husk ash (R).

**Figure 4 materials-16-07009-f004:**
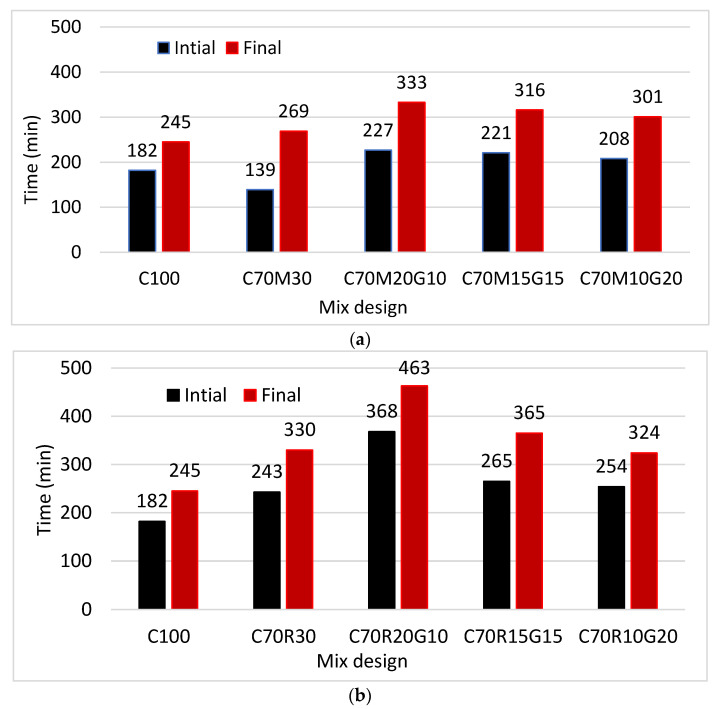
Setting time of the binder pastes with different contents of glass powder (G) for binders containing: (**a**) metakaolin (M) and (**b**) rice husk ash (R).

**Figure 5 materials-16-07009-f005:**
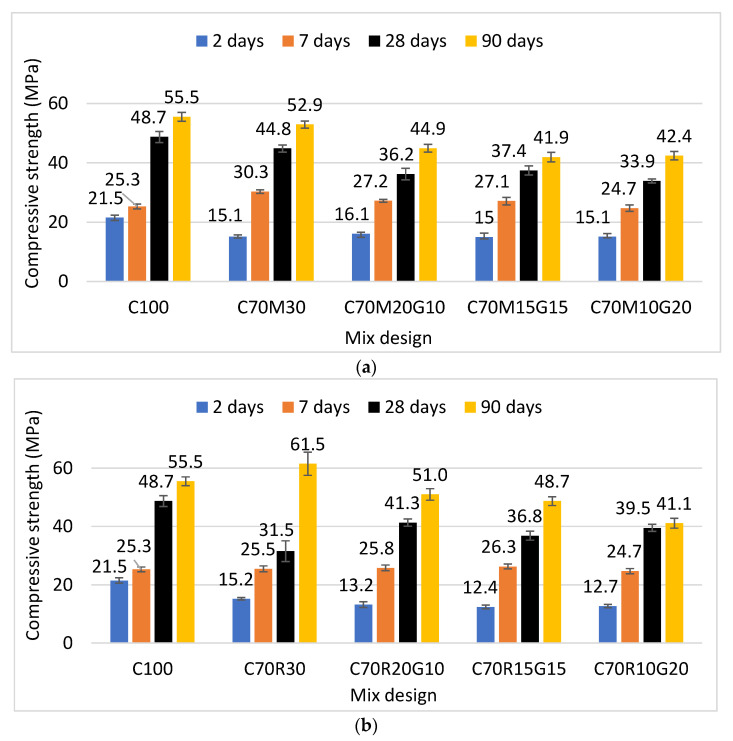
Compressive strength of binder mortars with different contents of glass powder (G) for binders containing: (**a**) metakaolin (M) and (**b**) rice husk ash (R).

**Figure 6 materials-16-07009-f006:**
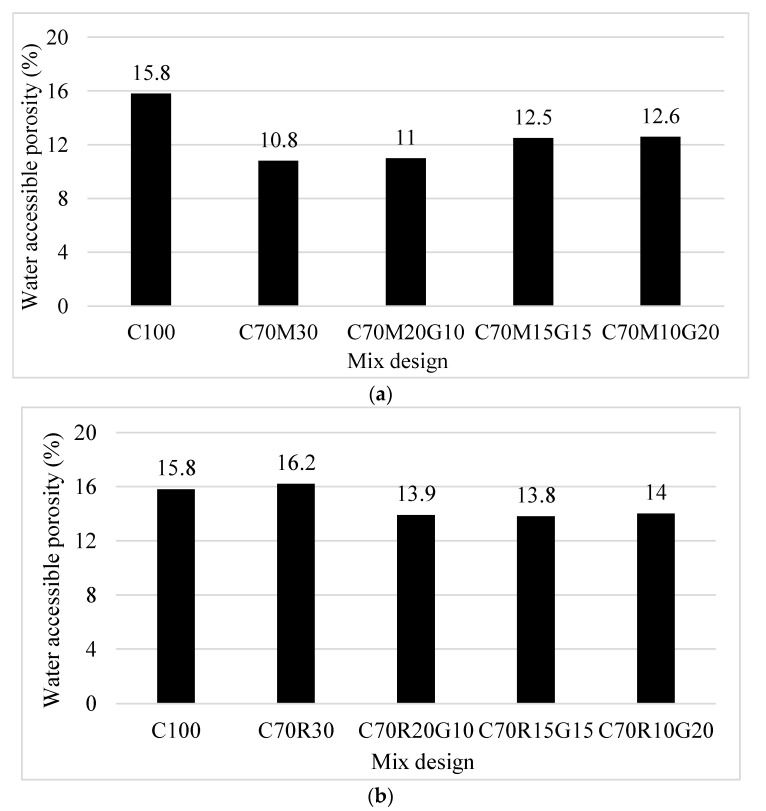
Water-accessible porosity of binder mortars with different contents of glass powder (G) for binders containing: (**a**) metakaolin (M) and (**b**) rice husk ash (R).

**Figure 7 materials-16-07009-f007:**
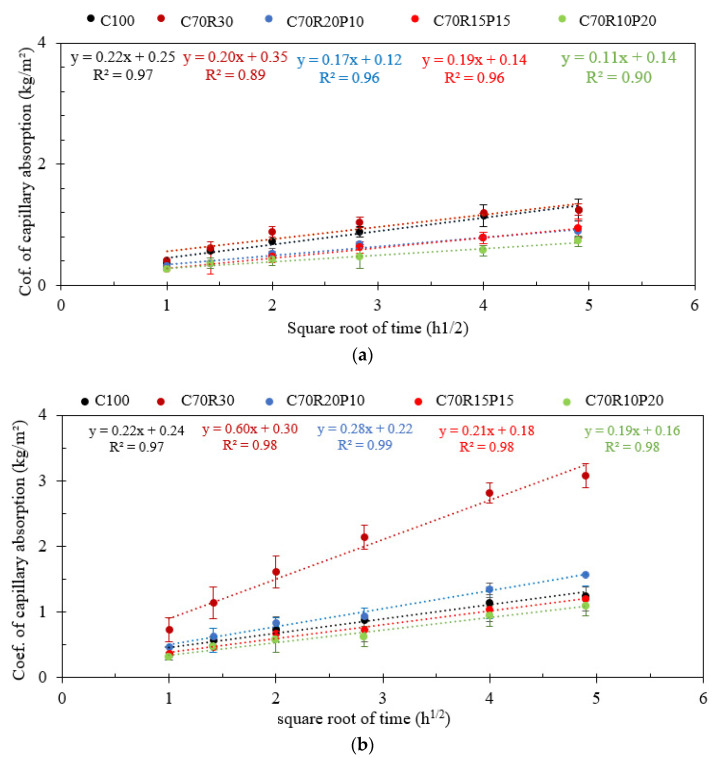
Capillary water absorption of binder mortars with different contents of glass powder (G) for binders containing: (**a**) metakaolin (M) and (**b**) rice husk ash (R).

**Figure 8 materials-16-07009-f008:**
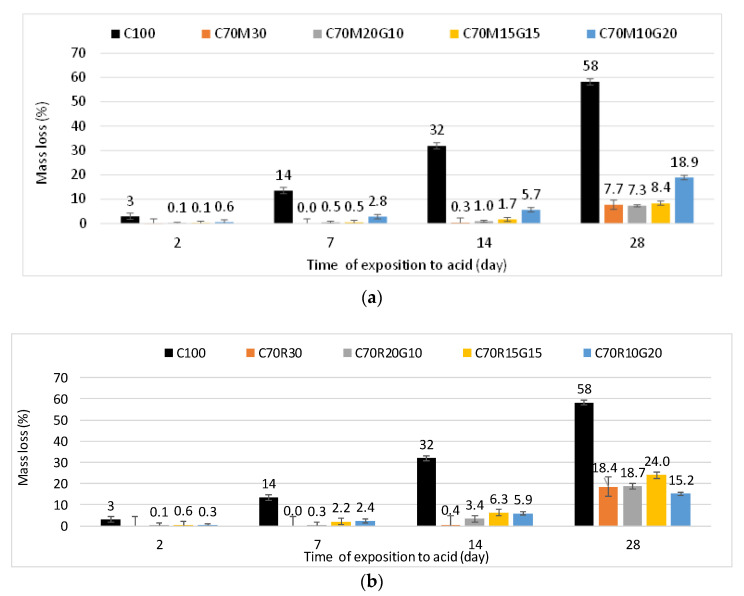
Evolution of mass loss due to acid attack of mortars with different contents of glass powder (G) for binders containing: (**a**) metakaolin (M) and (**b**) rice husk ash (R).

**Table 1 materials-16-07009-t001:** Chemical composition of materials.

Oxides (%)	SiO_2_	Al_2_O_3_	Fe_2_O_3_	CaO	MgO	Na_2_O	K_2_O	MnO_2_	TiO_2_	P_2_O_5_
MK ^a^	57.9	38.3	2.3	0.05	0.09	0.2	0.1	0.01	0.09	0.02
RHA ^b^	91.2	1.6	0.6	0.8	0.2	0	1.8	0.2	0.1	0.6
GP ^c^	68.1	0.9	0.6	14.5	1.8	12.2	0.8	-	-	-

^a^ [6]; ^b^ [17]; ^c^ [14].

**Table 2 materials-16-07009-t002:** Mix designs and compositions.

Designs (%)			CEM (g)	MK/RHA (g)	GP (g)	Sand (g)	Water (g)	Sp (g)
CEM	MK/RHA	GP						
100	0	0	450	0	0	1350	225	3.375
70	30	0	315	135	0	1350	225	3.375
70	20	10	315	90	45	1350	225	3.375
70	15	15	315	67.5	67.5	1350	225	3.375
70	10	20	315	45	90	1350	225	3.375

Note: These quantities are used to prepare three specimens (40 × 40 × 160 mm^3^) of mortars. The paste was made without sand and superplasticizer.

**Table 3 materials-16-07009-t003:** Evolution of the sorptivity and total water absorption of binder mortars with different contents of glass powder (G) for binders containing: (a) metakaolin (M) and (b) rice husk ash (R).

Samples	Sorptivity, 1–24 h (kg/m^2^.h^1/2^)	Correlation Coefficient, R^2^ (−)	Evolution of Pore Diameter (−)	Water-Accessible Porosity (%)
C100	0.22	0.97	*	15.8
C70M30	0.20	0.89	+	10.8
C70M20G10	0.17	0.96	−−	11.0
C70M15G15	0.19	0.96	−	12.5
C70M10G20	0.11	0.89	−−−−	12.6
C70R30	0.60	0.98	++++++++	16.2
C70R20G10	0.28	0.99	++	13.9
C70R15G15	0.21	0.98	*	13.8
C70R10G20	0.19	0.98	−	14.0

“*”: qualitative diameter of the control; “+”: increase and “−”: decrease in pore diameter.

## Data Availability

Not applicable.

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
