# Peer review of "Physico-Mechanical and Durability Characterization of Eco-Ternary Cementitious Binder Containing Calcined Clay/Rice Husk Ash and Recycled Glass Powder"

_materials, 2023, doi:10.3390/ma16217009_

Round 1
Reviewer 1 Report
The title has an error, which seems a bit unprofessional. Besides the title must be shorter and clearly show the content of the paper, which is not the case.
What is the novelty of the study concerning the literature survey? the literature review should be improved and highlight the novelty of the work. The work is easy to read and generally well explained but I didn´t see any novelty. it was a very limited study.
Please present the particle size and density of the sand used to produce mortars. and also provide SEM images of CEM, RHA, MTK and GP.
Why the chemical composition in Table 1 was obtained from other references? The authors did not perform? RHA, MTK and GP are not commercially available as far as I understand.
in Table 3 present the Water accessible porosity with same decimal places for all mortars
The conclusions are very limited since the work is also limited.
Reviewer 2 Report

can be improved more if more efforts can be paid.
Round 2
Reviewer 2 Report
1. I fully understood the substitution level now and maybe you can make it clearer to the reader. Instead of saying 20% of MK or RHA replaced by GP, it's better to express as what you replied to me.
2. I still cannot agree with the leaching part. Leaching normally refers to the hydration products leached out due to the surrounding water. Acid attacks normally lead to dissolution and decomposition of hydration products. It is suggested to modify the explanation you used for this. Besides, it’s better to cite some literature.
3. ‘Why MK led to an increase in the compressive strength after 7 days of curing seen in 70M30 compared to C100? You only explained the reason for a lower strength after 2 days and a slightly reduced strength after 90 days but did not mention this weird results’, but your Response :’There is no explanation that!’ I don’t understand your meaning.
4. ‘‘This shows that the mortars containing the RHA has lower resistance to acid attack than those with MK.’ I feel that this is not precise enough because after 28 days, 20% substitution level of RHA by GP displayed a smaller mass loss, which is 15.2% compared to the one with MK replace by GP. Again, your answer is not convincing.
The English expression still can be improved and some careless sentences or replies to the comments show that the authors should pay more attention on this.
